# Assessing the feasibility of large language models to identify top research priorities in enhanced external counterpulsation

**Shengkun Gai[1,2], Fangwan Huang[3], Xuanyun Liu[3], Ryan G. Benton[4], Glen M. Borchert[5], Jingshan Huang[6]\*, Xiuyu Leng [1,7]\***

**1** Department of Cardiology, The First Affiliated Hospital, Sun Yat-sen University, Guangzhou, China, **2** Department of Cardiology, Linfen People's Hospital, Linfen, Shanxi, China, **3** College of Computer and Data Science, Fuzhou University, Fuzhou, China, **4** School of Computing, University of South Alabama, Mobile, Alabama, United States of America, **5** College of Medicine, University of South Alabama, Mobile, Alabama, United States of America, **6** School of Computing and College of Medicine, University of South Alabama, Mobile, Alabama, United States of America, **7** NHC Key Laboratory of Assisted Circulation and Vascular Diseases (Sun Yat-sen University), Guangzhou, China

\* huang@southalabama.edu (JH); lengxy@mail.sysu.edu.cn (XL)

## Abstract

Enhanced External Counterpulsation (EECP), as a non-invasive, cost-effective, and efficient adjunctive circulatory technique, has been widely applied in in the cardiovascular field. Numerous studies and clinical observations have confirmed the obvious advantages of EECP in promoting blood flow perfusion to vital organs such as the heart, brain, and kidneys. However, many potential mechanisms of EECP remain insufficiently validated, necessitating researchers to dedicate substantial time and effort to in-depth investigations. In this work, large language models (such as ChatGPT and Ernie Bot) were used to identify top research priorities in five key topics in the field of EECP: mechanisms, device improvements, cardiovascular applications, neurological applications, and other applications. After generating specific research priorities in each domain through language models, a panel of nine experienced EECP experts was invited to independently evaluate and score them based on four parameters: relevance, originality, clarity, and specificity. Notably, high average and median scores for these evaluation parameters were obtained, indicating a strong endorsement from experts in the EECP field. This study preliminarily suggests that large language models like ChatGPT and Ernie Bot could serve as powerful tools for identifying and prioritizing research priorities in the EECP domain.

## 1. Introduction

Enhanced External Counterpulsation (EECP) is a non-invasive adjunctive circulatory technique that inflates and deflates cuffs wrapped around the limbs and buttocks in sync with the cardiac cycle under electrocardiographic gating control. EECP has been clinically demonstrated to significantly improve organ perfusion, regulate endothelial function, combat coronary artery atherosclerosis, treat complications of diabetes and sudden sensorineural hearing loss, among other benefits [1–3]. Although many evidences suggest there is a great deal of untapped potential for external counterpulsation, traditional approaches to identifying

**Data availability statement:** All relevant data are within the manuscript and its Supporting Information files.

**Funding:** The author(s) received no specific funding for this work.

research priorities for EECP mainly rely on expert opinion and consensus building which are often labor-intensive and biased. In recent years, natural language processing (NLP) technology [4] has been increasingly recognized as a new means of identifying research priorities. Large language models (LLMs) such as ChatGPT [5] and Ernie Bot [6], which are trained on extensive text data, possess the ability to understand human-like language and have demonstrated significant potential in proposing and prioritizing research priorities [7]. In the medical domain, LLMs have shown promising results in various tasks, including disease diagnosis, medical record automation, literature retrieval, and patient education [8]. Adi Lahat et al. assessed the effectiveness of ChatGPT in generating research questions within gastroenterology and concluded that ChatGPT could be used to produce high-quality research inquiries [9]. Building on the recognition of NLP technology and the potential of large language models like ChatGPT and Ernie Bot to identify research priorities, their effectiveness in determining primary research priorities related to EECP technology were specifically evaluated in this work. Five key areas were examined: mechanisms, device enhancements, cardiovascular applications, neurological applications, and other applications. Utilizing ChatGPT and Ernie Bot, specific research priorities in these domains were generated, after which they were reviewed by experienced EECP experts and then rated to assess their relevance and importance.

## 2. Related work

Large language models have shown broad applicability in entertainment, education, and customer service, but their potential in the medical field remains largely untapped. Given the high standards for information quality and communication reliability in medicine, the application of large language models requires careful consideration. In recent years, scholars have begun to explore the use of large language models in medicine, yielding promising results. In the field of cardiology, Gala et al. [10] believed that LLMs can be utilized to analyze a large number of academy papers and medical record resources to help clinicians keep up with the latest advances in cardiology. Nevertheless, they also pointed to the limitations of LLMs in explaining cultural or emotional factors that may influence medical practice. Cascella et al. [11] explored the reasoning abilities of ChatGPT on public health topics. Through a question-and-answer session, ChatGPT listed four possible research topics. While some of the responses of ChatGPT may be stereotyped and depend on the prompts, it can be used to summarize the scientific literature and generate new research hypotheses. Additionally, George et al. [12] proposed that large language models could serve as a supplementary resource to traditional medical tools, improving the efficiency and productivity of medical practices. Unfortunately, these studies do not provide a quantitative assessment of the ability of LLMs to identify medical research priorities.

Importantly, in order to assess the effectiveness of LLMs in the medical domain, it is essential to conduct statistical analyses on numerical results obtained from experiments and/or surveys. In evaluating the pertinent literature on LLMs, Tang et al. [13] invited field experts to assess the summary quality of LLMs by using a five-point Likert scale along four dimensions: coherence, factual consistency, comprehensiveness, and harmfulness. Man-Whitney U test was used to assess the differences in response between GPT-3.5 and ChatGPT. Michael et al. [14] employed average scoring and fixed-effects consistency to calculate the Intraclass Correlation Coefficient (ICC), investigating the potential application of artificial intelligence-based LLMs in the realm of medical ethics. Similarly, Dave et al. [15] utilized Pearson and Spearman coefficients to juxtapose the assessment outcomes of large language models against the evaluations of medical professionals, thereby further substantiating their dependability. Furthermore, besides correlation analysis, similarity metrics are frequently utilized to gauge the efficacy of LLMs. For example, in 2024, Sebastian et al. [16] evaluated the pairwise accuracy between LLMs and human assessments by

analyzing the cosine similarity matrix. In measuring factual knowledge within LLMs, Pezesh-kpour [17] successfully utilized Kullback-Leibler (KL) divergence to analyze the predictive probability distributions of the model before and after instilling target knowledge. In investigating bias issues within large pre-trained language models, Guo et al. [18] used the Jensen-Shannon (JS) divergence to measure the consistency between different demographic distributions, offering a robust tool for reducing human-like biases and unwanted societal stereotypes. JS divergence is an improved version of KL divergence, whereas the KL scatter is asymmetric, making the JS scatter more accurate in identifying similarities.

## 3. Methods

### 3.1. Research priorities

ChatGPT (based on GPT-3.5) and Ernie Bot 3.5 to generate research priorities in five key topics (Tables 1 and 2, respectively) pertaining were leveraged to EECP mechanisms [1,19], structural enhancements, applications in cardiovascular domains [3,20,21], neurological applications [22, 23], and other applications [3,24,25].

### 3.2. Expert evaluation

The expert evaluation panel was comprised of nine highly experienced EECP specialists as evidenced by panelists having authored an average of twenty relevant research publications in the field. They gained their expertise through clinical practice and made significant contributions to academic research, and experts have published at least five scholarly articles related to EECP. Furthermore, they have actively contributed to the development of guidelines in the EECP field. Panelists reviewed and assessed the inquiries presented by ChatGPT and Ernie Bot independently. Experts rated five priorities on four parameters (relevance, originality, clarity, and specificity) using a 1–5 scale with 5 representing the highest score. The a priori relationships generated by ChatGPT and Ernie Bot were then compared to current EECP research queries identified through a manual literature review. Importantly, in order to ensure the objectivity and relevance of responses, ChatGPT and Ernie Bot were instructed to treat each key topic as an independent query, thereby eliminating potential biases that may have existed in previous conversations.

## 4. Statistical analysis

Data were collected and analyzed using standard statistical methods, and all statistical analyses were conducted using IBM SPSS Statistics version 25 and Python 3.10. Initially, descriptive statistical methods were employed to provide a summary of the data, including measures such as mean, standard deviation (SD), and median. Afterwards, "divergence" was adopted to assess the similarity between ratings provided by experts in EECP and queries generated by two large language models. In the realm of data mining, JS divergence was computed to evaluate the similarity of ratings among evaluators using a rating table structured with evaluators as column attributes. JS divergence values from 0 to 1, with smaller values indicating greater similarity between ratings. Additionally, Spearman's rank correlation coefficient and Kendall's τ coefficient were also used to evaluate pairwise correlations between parameters. Positive coefficients indicate a positive correlation, while negative coefficients imply a negative correlation. The closer the coefficient is to 1 the stronger the correlation.

## 5. Results

The statistical analysis shows high reliability for the questionnaires assessing ChatGPT and Ernie Bot, with Cronbach's alpha coefficients of 0.978 and 0.971, respectively. Both coefficients

**Table 1. ChatGPT-generated research priorities on five key topics in the field of EECP research.**

| Topic | ChatGPT-generated Research Priorities |
|---|---|
| Mechanisms | 1. Integrated Insights into EECP Effects: Unveiling Hemodynamic, Neuroendocrine, and Inflammatory Responses and Their Implications for Cardiovascular and Systemic Health. <br> 2. How does EECP compare to other treatment modalities such as pharmacotherapy, invasive procedures, or lifestyle interventions? <br> 3. Optimizing EECP Treatment: Investigating Timing and Frequency for Varied Patient Populations and Clinical Conditions. <br> 4. Refining Patient Selection for EECP: Optimizing Outcomes Based on Demographics, Comorbidities, and Treatment Responsiveness. <br> 5. Assessing the Long-Term Impact of EECP Therapy: Prospective Studies on Cardiovascular Outcomes, Quality of Life, and Healthcare Utilization. |
| Device improvements | 1. Revolutionizing EECP Treatment: Evaluating Compact and Portable Devices for Home-Based and Ambulatory Care, Integrating Technological Advancements for Enhanced Efficacy. <br> 2. Advancements in EECP Technology: Improving Treatment Efficacy and Patient Experience through Compact and Portable Devices, Automated Pressure Control, and Real-Time Monitoring. <br> 3. Enhancing Patient Experience in EECP Treatment: Investigating Ergonomic Enhancements for Adjustable Seating, Cushioning, and Noise Reduction. <br> 4. Integrating EECP Devices with Telemedicine and Mobile Health: Enhancing Remote Monitoring and Patient Care. <br> 5. Optimizing EECP Treatment: Investigating Safety Mechanisms for Enhanced Patient Security. |
| In the field of heart disease | 1. Unlocking Vascular Relaxation: Investigating Signaling Pathways in EECP Treatment through eNOS, cGMP, and Vasodilatory Factors. <br> 2. Unveiling the Future of Cardiac Health: Investigating EECP Treatment Integration with Telemedicine and Mobile Health for Remote Monitoring. <br> 3. Longitudinal Assessment of EECP Impact on Endothelial Function, Progenitor Cell Activity, and Vascular Inflammation: Implications for Cardiovascular Disease Prevention and Management. <br> 4. Unraveling Dynamic Hemodynamic Responses: Investigating Cardiac Output, Systemic Vascular Resistance Changes, and Implications for Non-Pharmacological Heart Failure Management in EECP Treatment. <br> 5. Examining Autonomic Regulation: HRV Parameters, Sympathetic-Parasympathetic Balance, and Baroreflex Sensitivity Dynamics Following EECP Treatment. |
| Applications in the field of neurology | 1. Neuroimaging Insights into Structural and Functional Changes Induced by EECP: Implications for Neural Repair and Recovery Processes. <br> 2. Synergistic Approaches to Neurological Rehabilitation: Investigating the Combined Effects of EECP with Physical, Occupational, and Cognitive Interventions on Motor and Cognitive Recovery Post-Stroke. <br> 3. Enhancing Neurological Recovery: Assessing the Impact of EECP on Cognitive Function, Neuroplasticity, and Rehabilitation Potential Following Traumatic Brain Injury. <br> 4. Exploring the Neurotrophic and Neuroregenerative Potential of EECP: Implications for Disease Pathology, Cognitive Decline, and Motor Symptoms. <br> 5. The Impact of Enhanced External Counterpulsation on Neurovascular Coupling and Cerebral Perfusion: Investigating Mechanisms and Potential Therapeutic Applications in Neurovascular Disorders. |
| Applications in other fields | 1. Unveiling the Metabolic Impact of EECP: Exploring Effects on Insulin Sensitivity, Lipid Profiles, and Inflammatory Markers for Managing Metabolic Syndrome and Cardiovascular Risk Factors. <br> 2. Optimizing Orthopedic Wellness: Evaluating the Impact of EECP on Joint Function, Muscle Strength, and Pain Management in Osteoarthritis, Rheumatoid Arthritis, and Musculoskeletal Injuries. <br> 3. Enhancing Sleep Quality with EECP: Exploring Effects on Sleep Architecture, Latency, and Efficiency as a Non-Pharmacological Intervention for Sleep Disorders <br> 4. Renal Rejuvenation: Investigating the Effects of EECP on Renal Blood Flow, Glomerular Filtration Rate, and Kidney Injury Markers for Managing Renal Artery Stenosis and Acute Kidney Injury. <br> 5. Utilizing Enhanced External Counterpulsation to Enhance Cancer Treatment Outcomes: Investigating Chemotherapy Delivery, Cardiotoxicity Reduction, and Treatment Efficacy Enhancement. |

exceed the 0.8 threshold, indicating strong survey reliability. This suggests that the questionnaires effectively reflect the proficiency of ChatGPT and Ernie Bot in determining research priorities for EECP.

Based on this, the study conducted data analysis on the ratings provided by the 9 evaluators from three perspectives: (1) descriptive statistics; (2) similarity of ratings among evaluators; and (3) rank correlation of evaluation metrics. The data analysis tools utilized were IBM SPSS Statistics Version 25 and Python 3.10.

## 5.1 Descriptive statistics

Three score tables for each large language model were constructed, featuring evaluation metrics, evaluators, and topics as column attributes. For example, in the score table with five topics as column attributes, each column represents the scores from nine evaluators on four evaluation indicators for five research priorities within a specific topic. As shown in Tables 3–5, the results were derived from descriptive statistics applied to these three score tables. Since the mean and standard deviation have been commonly used to describe normal or approximately normal distributions, the quartiles in Tables 3–5 were considered to accurately reflect potential non-normal distributions. It is believed that the combination of mean/standard deviation and quartiles effectively reduces the impact of extreme values that may not fully represent the actual situation. From Table 3, it is clear that the two large language models excel in relevance, with originality following closely behind. In-depth descriptive statistical analyses of evaluation metrics are presented in Tables 3–5. The major models performed best in relevance, with originality close behind. Although originality exhibited the largest standard deviation, suggesting significant variation in expert opinions regarding originality, clarity demonstrated the smallest standard deviation, indicating minimal fluctuations in scores for each question. Additionally, variations in performance between the two models (ChatGPT and Ernie Bot) across different evaluation metrics and topics can be observed. Concerning relevance, Ernie Bot's average score slightly exceeds ChatGPT's, suggesting a slight advantage in addressing user-related questions, although this was not statistically significant. In terms of originality, ChatGPT's score was slightly less than Ernie Bot's, with a higher fluctuation in scoring standard deviation, indicating some disagreement among experts regarding the originality of ChatGPT's queries. Both models demonstrate similar performance in clarity and specificity, indicating their similarity in providing clear and specific answers. Results of scores from EECP experts for all priorities are visually presented in Fig 1 with the outermost rings corresponding to the highest score of 5 and inner rings indicating lower scores.

Table 4 presents the scores given by different raters for the ChatGPT and Ernie Bot models. The analysis shows that in the evaluations of most raters, ChatGPT and Ernie Bot have similar average scores indicating a certain level of competitiveness in overall performance. However, it is worth noting that in the ratings of Rater3 and Rater4, Ernie Bot's average score was clearly higher than ChatGPT's, reflecting a more outstanding performance of Ernie Bot from the perspectives of these two raters. In terms of score stability, there were differences between the two models among different raters. Specifically, in the evaluations of Rater3 and Rater4, Ernie Bot had a lower standard deviation, indicating more stable scores and consistent performance. Conversely, Rater8's Ernie Bot scores demonstrated significantly higher standard deviation. In contrast, although overall score stability was slightly inferior to Ernie Bot's performance for a subset of raters, ChatGPT's standard deviation among multiple raters was relatively more consistent. These differences in evaluation may stem from personal preferences, evaluation criteria, and model performance across different topics.

**Table 2. ERNIE bot-generated research priorities on five key topics in the field of EECP research.**

| Topic | ERNIE Bot-generated Research Priorities |
|---|---|
| Mechanisms | 1. Study on the impact of enhanced external counterpulsation on hemodynamics and collateral vessel development.<br>2. Molecular mechanisms regulating endothelial cell function and smooth muscle cell behavior in EECP.<br>3. Exploration of the mechanism of enhanced external counterpulsation based on genomics.<br>4. Application of real-time imaging techniques in assessing the therapeutic effects of enhanced external counterpulsation.<br>5. Comparative study of the efficacy of enhanced external counterpulsation and traditional drug therapy in cardiovascular and cerebrovascular diseases |
| Device improvements | 1. Application of intelligent perception and adaptive adjustment technology in enhanced external counterpulsation machines.<br>2. Optimization design of enhanced external counterpulsation machines based on biomechanical simulation.<br>3. Research and application of new materials and technologies in enhanced external counterpulsation machines.<br>4. Development and preliminary evaluation of wearable enhanced external counterpulsation devices.<br>5. Research on multimodal feedback and control systems in enhanced external counterpulsation machines. |
| In the field of heart disease | 1. Effect prediction and optimization of enhanced external counterpulsation in cardiovascular treatment based on big data analysis.<br>2. Exploration of the potential of enhanced external counterpulsation in post-heart transplant rehabilitation.<br>3. Mechanism study using enhanced external counterpulsation to promote heart regeneration and repair.<br>4. Exploration of personalized treatment strategies using enhanced external counterpulsation in cardiovascular medicine.<br>5. Long-term monitoring and management research in cardiovascular medicine using wearable enhanced external counterpulsation technology. |
| Applications in the field of neurology | 1. Mechanism of enhanced external counterpulsation in cerebral blood flow reconstruction in ischemic stroke patients.<br>2. Evaluation of the cognitive function improvement effect of enhanced external counterpulsation based on neuroimaging.<br>3. Promotion of brain function reshaping in neurorehabilitation through enhanced external counterpulsation.<br>4. Prospective study using enhanced external counterpulsation to improve the quality of life in neuropathic pain patients.<br>5. Synchronization study of brain blood flow and neural activity using enhanced external counterpulsation technology. |
| Applications in other fields | 1. Exploration of the mechanism of skin regeneration and wound healing using enhanced external counterpulsation technology.<br>2. Application of enhanced external counterpulsation in improving renal function in chronic kidney disease patients.<br>3. Promotion of healing and vascular regeneration in diabetic foot ulcers using enhanced external counterpulsation.<br>4. Potential study of enhanced external counterpulsation in improving joint blood flow and inflammation control in rheumatoid arthritis patients.<br>5. Role of enhanced external counterpulsation in promoting skin regeneration and reducing scar formation after burns. |

**Table 3. Descriptive statistics of evaluation metrics.**

| Evaluation Metric | Model | Mean | Standard Deviation | Lower Quartile | Median | Upper Quartile |
|---|---|---|---|---|---|---|
| **relevance** | ChatGPT | 3.88 | 0.99 | 3.00 | 4.00 | 5.00 |
| | Ernie Bot | 4.04 | 1.06 | 4.00 | 4.00 | 5.00 |
| **originality** | ChatGPT | 3.70 | 1.05 | 3.00 | 4.00 | 5.00 |
| | Ernie Bot | 3.88 | 1.10 | 3.00 | 4.00 | 5.00 |
| **clarity** | ChatGPT | 3.52 | 0.92 | 3.00 | 4.00 | 4.00 |
| | Ernie Bot | 3.56 | 0.95 | 3.00 | 4.00 | 4.00 |
| **specificity** | ChatGPT | 3.40 | 0.93 | 3.00 | 3.00 | 4.00 |
| | Ernie Bot | 3.46 | 0.97 | 3.00 | 4.00 | 4.00 |

In all topics (Table 5), Ernie Bot consistently received higher average scores than ChatGPT, suggesting a relative advantage in overall performance. Although their performances in terms of median scores were similar, Ernie Bot achieved an upper quartile score of 5.00 in specific topics such as mechanisms, device improvements and applications in neurology, indicating higher recognition in these areas. Meanwhile ChatGPT's standard deviation across multiple topics was slightly lower than Ernie Bot's, suggesting relatively better score stability. However, this difference was not significant. Notably, clear domain-specific differences were observed, while Ernie Bot's average score significantly surpassed ChatGPT's in structural improvements and applications in neurology domains, ChatGPT demonstrated superior performance in other domains.

## 5.2 Similarity of raters' scores

Regarding the similarity of raters' scores, the JS divergence of scores between each pair of raters for ChatGPT and Ernie Bot was calculated (Fig 2). The results indicate that the JS divergence range of scores for ChatGPT is [0, 0.102], while for Ernie Bot, it is [0, 0.148]. Since a smaller JS divergence value indicates higher similarity, it can be concluded that the evaluations of these two large language models by raters exhibit relatively high consistency. It is worth noting that, for both ChatGPT and Ernie Bot, the similarity of scores between rater 8 and other raters is the lowest. From Fig 1, it is evident that the scores given by rater 8 are significantly lower than those given by other raters. Further analysis of the data in Table 4 reveals that the average scores given by rater 8 for both ChatGPT and Ernie Bot are the lowest (2.20 and 2.44 respectively). Besides, they have the highest standard deviations (0.80 and 1.21 respectively). Excluding the influence of rater 8's scores, the upper limit of the JS divergence of scores for ChatGPT would decrease from 0.102 to 0.052, and from 0.148 to 0.063 for Ernie Bot.

## 5.3 Correlation of evaluation metrics

In terms of the correlation of evaluation metrics, we calculated both the Spearman [26] and Kendall [27] coefficients between pairs of evaluation metrics in the scoring results for ChatGPT and Ernie Bot (see Tables 6 and 7). These analyses passed significance tests, with all p-values below 0.01 indicating a significant positive correlation between relevance, originality, clarity, and specificity. This implies that when evaluating these two models, the score trends among these metrics were consistent, demonstrating high consistency and reliability. That said, ChatGPT exhibited a lower correlation between originality and relevance, while Ernie Bot showed a lower correlation in the analysis of specificity and relevance. The clarity of both models was highly correlated with relevance and/or specificity.

## 6. Discussion

Here, the ability of ChatGPT and Ernie Bot was evaluated to generate research priorities in the field of EECP, covering mechanisms, structural improvements, applications in cardiology, applications in neurology, and applications in other fields. Both models demonstrated significant potential in consistently generating relevant and clear research priorities, which could offer valuable new tools for EECP research. Both scored relatively low in specificity, possibly due to limitations in handling domain-specific knowledge, indicating a need for improvement in accuracy and precision. To enhance their performance, fine-tuning with domain-specific data and expert knowledge will likely be required. While both models lacked originality in their responses, relying heavily on learned information and language patterns, future research should focus on enhancing their creativity to generate more unique research questions in the EECP field.

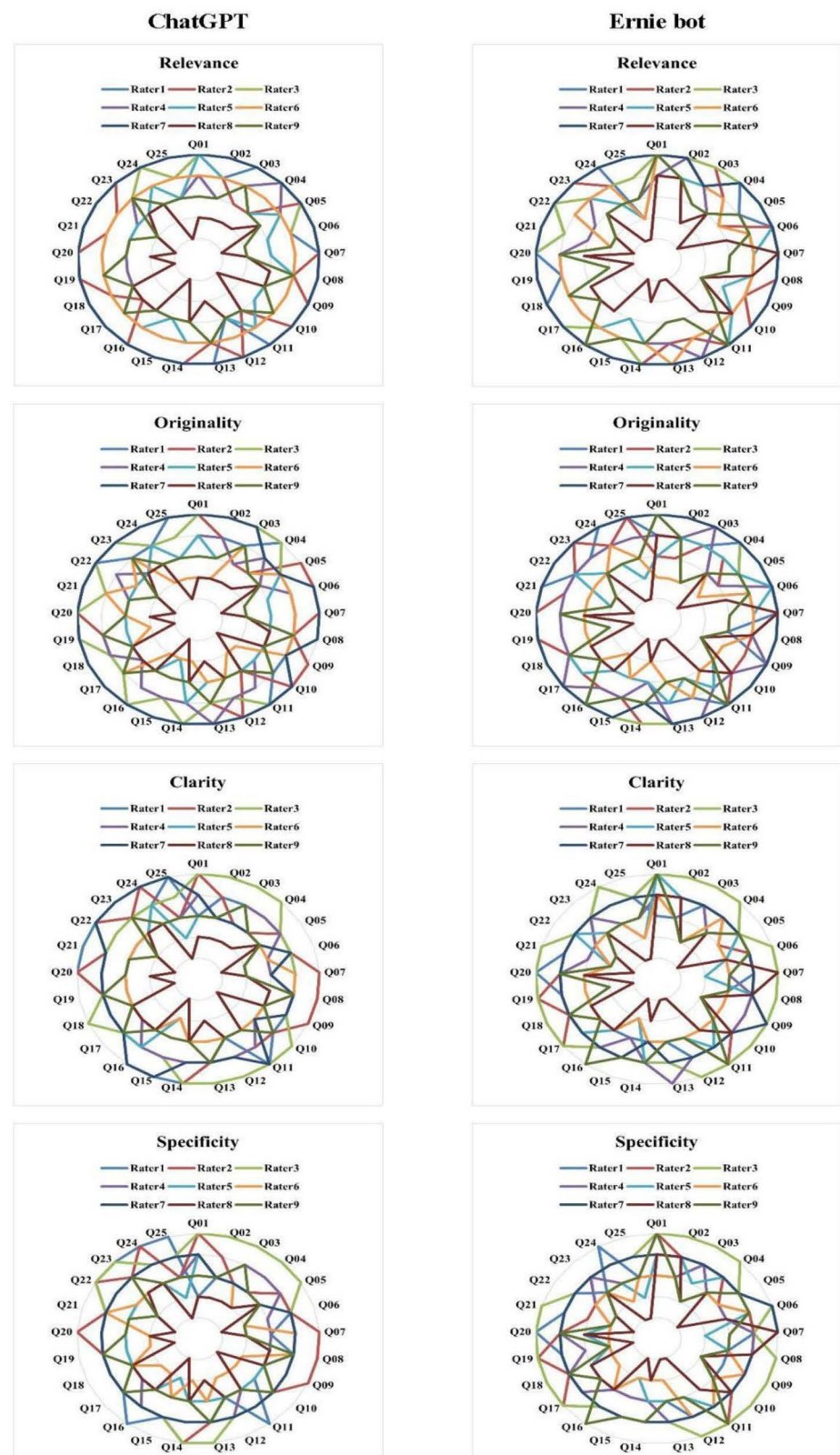

**Fig 1. Ratings of 25 research focal points by nine evaluators based on four criteria.**

**Table 4. Descriptive statistics of evaluator.**

| Evaluator | Model | Mean | Standard Deviation | Lower Quartile | Median | Upper Quartile |
|-----------|-------|------|--------------------|----------------|--------|----------------|
| **Rater1** | ChatGPT | 4.34 | 0.71 | 4.00 | 4.00 | 5.00 |
| | Ernie Bot | 4.10 | 0.78 | 4.00 | 4.00 | 5.00 |
| **Rater2** | ChatGPT | 4.30 | 0.70 | 4.00 | 4.00 | 5.00 |
| | Ernie Bot | 4.20 | 0.67 | 4.00 | 4.00 | 5.00 |
| **Rater3** | ChatGPT | 4.37 | 0.49 | 4.00 | 4.00 | 5.00 |
| | Ernie Bot | 4.75 | 0.44 | 4.25 | 5.00 | 5.00 |
| **Rater4** | ChatGPT | 3.50 | 0.54 | 3.00 | 3.00 | 4.00 |
| | Ernie Bot | 3.81 | 0.61 | 3.00 | 4.00 | 4.00 |
| **Rater5** | ChatGPT | 3.16 | 0.58 | 3.00 | 3.00 | 3.00 |
| | Ernie Bot | 3.37 | 0.86 | 3.00 | 3.00 | 4.00 |
| **Rater6** | ChatGPT | 3.17 | 0.77 | 3.00 | 3.00 | 4.00 |
| | Ernie Bot | 3.14 | 0.79 | 3.00 | 3.00 | 4.00 |
| **Rater7** | ChatGPT | 4.44 | 0.67 | 4.00 | 5.00 | 5.00 |
| | Ernie Bot | 4.50 | 0.52 | 4.00 | 5.00 | 5.00 |
| **Rater8** | ChatGPT | 2.20 | 0.80 | 2.00 | 2.00 | 3.00 |
| | Ernie Bot | 2.44 | 1.21 | 1.00 | 3.00 | 3.00 |
| **Rater9** | ChatGPT | 3.16 | 0.61 | 3.00 | 3.00 | 4.00 |
| | Ernie Bot | 3.31 | 0.92 | 3.00 | 3.00 | 4.00 |

**Table 5. Descriptive statistics of topic.**

| topic | Model | Mean | Standard Deviation | Lower Quartile | Median | Upper Quartile |
|-------|-------|------|--------------------|----------------|--------|----------------|
| **mechanisms** | ChatGPT | 3.63 | 0.95 | 3.00 | 4.00 | 4.00 |
| | Ernie Bot | 3.83 | 0.98 | 3.00 | 4.00 | 5.00 |
| **device improvements** | ChatGPT | 3.52 | 1.03 | 3.00 | 4.00 | 4.00 |
| | Ernie Bot | 3.94 | 0.90 | 3.00 | 4.00 | 5.00 |
| **in the field of heart disease** | ChatGPT | 3.69 | 0.98 | 3.00 | 4.00 | 4.00 |
| | Ernie Bot | 3.68 | 1.12 | 3.00 | 4.00 | 4.75 |
| **in the field of neurology** | ChatGPT | 3.67 | 0.91 | 3.00 | 4.00 | 4.00 |
| | Ernie Bot | 3.85 | 0.99 | 3.00 | 4.00 | 5.00 |
| **the other field** | ChatGPT | 3.62 | 1.06 | 3.00 | 4.00 | 4.00 |
| | Ernie Bot | 3.38 | 1.14 | 3.00 | 3.00 | 4.00 |

Notably, the performances of Ernie Bot and ChatGPT, two prominent language systems were compared. Ernie Bot demonstrated a slight but definitive advantage in terms of relevance, possibly due to its more precise semantic understanding and higher matching with user needs. In terms of originality, ChatGPT scored slightly lower with more fluctuation, indicating some disagreement among evaluators regarding its ability to offer novel and unique perspectives. This variance might stem from differences in the models' performance across different contexts or from evaluators' subjective criteria, such as their acceptance of research priorities that challenge existing cognitive frameworks or their willingness to explore unknown areas of study. In contrast, Ernie Bot received more consistent recognition for its originality, likely due to its more flexible and innovative thinking patterns. Regarding clarity and specificity, both models performed equally well, demonstrating high levels of proficiency. This suggests that they excel in providing clear, understandable responses and specific, detailed explanations,

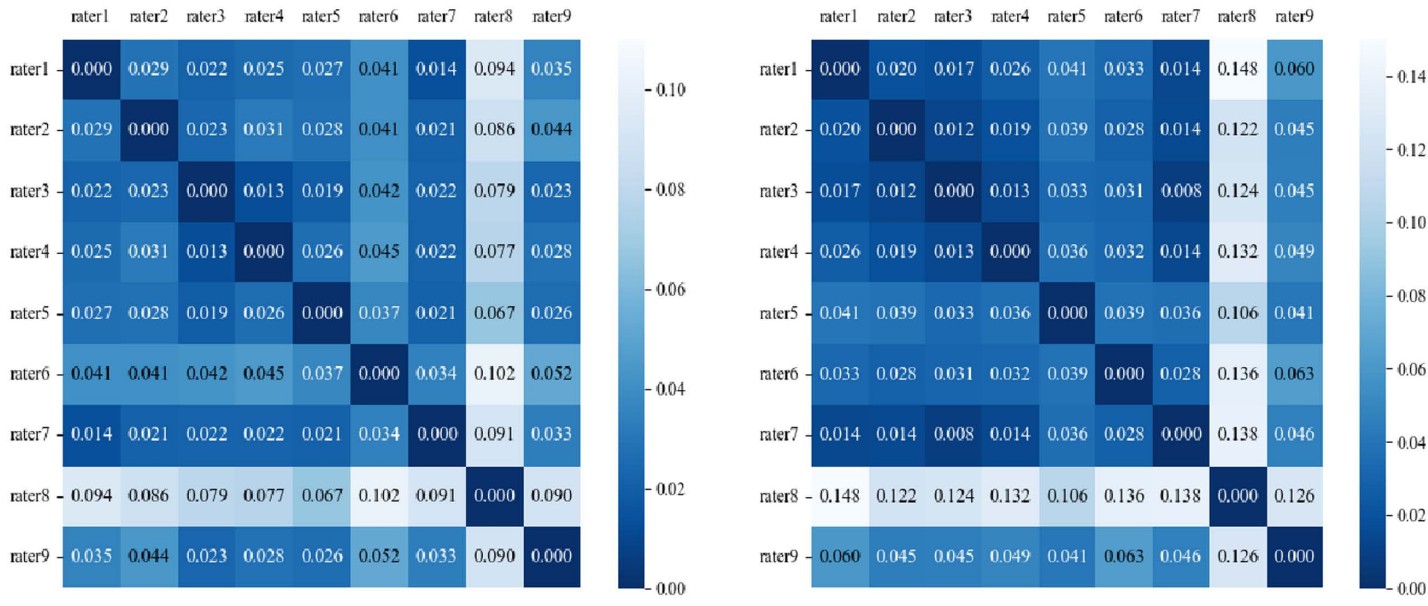

(a)ChatGPT                    (b)Ernie Bot

**Fig 2. JS divergence heat map depicting the similarity of ratings between pairs of evaluators.**

which are equally important for large language models as users often expect answers that are both clear and specific to better understand and apply the provided information.

From the evaluators' perspective, most evaluators held similar views on the performance of the two models. However, in certain specific cases, such as Rater3 and Rater4, Ernie Bot received higher scores. Additionally, as compared to other raters, Rater8's scores were significantly lower and deviated more substantially, and exclusion of Rater8 increased the performance of both models.

In certain specific topics such as mechanisms, applications in neurology, and cardiovascular applications, Ernie Bot performed better whereas ChatGPT's performance slightly

**Table 6. Rank correlation coefficients between evaluation metrics (ChatGPT).**

| Spearman's coefficient | relevance | originality | clarity | specificity |
|---|---|---|---|---|
| relevance | 1 | 0.778** | 0.722** | 0.670** |
| originality | 0.778** | 1 | 0.780** | 0.772** |
| clarity | 0.722** | 0.780** | 1 | 0.883** |
| specificity | 0.670** | 0.772** | 0.883** | 1 |
| Kendall's coefficient | relevance | originality | clarity | specificity |
| relevance | 1 | 0.713** | 0.667** | 0.605** |
| originality | 0.713** | 1 | 0.726** | 0.711** |
| clarity | 0.667** | 0.726** | 1 | 0.840** |
| specificity | 0.605** | 0.711** | 0.840** | 1 |

Note:

**Significance at the 0.01 level (two-tailed).

Table 7. Rank correlation coefficients between evaluation metrics (Ernie Bot).

| Spearman's coefficient | relevance | originality | clarity | specificity |
|---|---|---|---|---|
| relevance | 1 | 0.692** | 0.695** | 0.708** |
| originality | 0.692** | 1 | 0.740** | 0.769** |
| clarity | 0.695** | 0.740** | 1 | 0.876** |
| specificity | 0.708** | 0.769** | 0.876** | 1 |
| Kendall's coefficient | relevance | originality | clarity | specificity |
| relevance | 1 | 0.643** | 0.646** | 0.650** |
| originality | 0.643** | 1 | 0.686** | 0.707** |
| clarity | 0.646** | 0.686** | 1 | 0.846** |
| specificity | 0.650** | 0.707** | 0.846** | 1 |

Note:

**Significance at the 0.01 level (two-tailed).

surpassed that of Ernie Bot in others, indicating that each model has its strengths and weaknesses in different domains and application scenarios.

Consequently, future research should be performed explore how to effectively integrate the strengths of both models to improve the performance and efficacy of large language models in real-world applications.

Our study applied ChatGPT and ERNIE Bot in the field of EECP to identify high-quality research priorities for the first time. It also offers a cross-disciplinary examination of the potential applications of EECP in neurology, metabolism, orthopedics, nephrology, and other areas. Furthermore, combining expert evaluations with statistical analysis enhances the scientific rigor and accuracy of our findings. This novel approach not only advances the development and refinement of EECP technology but also opens up new possibilities for patient treatment.

## 7. Limitations

Although this study presents promising outcomes, there are also some limitations in this study. Firstly, the expert panels involved may not fully represent the broader research community, which could have influenced the evaluation results. Secondly, the use of subjective ratings may introduce bias and variability in assessing the performance of ChatGPT and Ernie Bot. Lastly, the models may not have access to the latest biomedical literature, which could affect the quality of question generation. If this is the case, integrating domain-specific APIs with up-to-date information could enhance research quality. For future work, key directions include improving expert panel representation, optimizing large language models with more domain-specific training data, enhancing data transparency, applying more robust statistical methods, and fostering interdisciplinary collaboration. These efforts aim to address the identified limitations and promote innovation and advancement in EECP research.

## 8. Conclusion

Overall, this assessment of ChatGPT and Ernie Bot as generators of research priorities for Enhanced External Counterpulsation (mechanisms, device improvements, applications in cardiovascular medicine, applications in neurology, and applications in other non-cardiovascular and non-neurological fields) produced some promising results. Both models have demonstrated the capacity to generate high-quality research priorities in these areas, indicating their potential value as tools to drive research not only in EECP but also in broader medical fields

through streamlining the process of identifying crucial research priorities and thereby save considerable time and effort. While there is room for improvement in terms of specificity and originality, both models have shown a capability to produce diverse, relevant, and coherent research priorities, likely aiding advancements in EECP research. Each model has its strengths in various domains and application scenarios, and further exploration could focus on leveraging these strengths to enhance the overall effectiveness of large language models in practical settings. In conclusion, our findings suggest that ChatGPT and Ernie Bot are poised to become valuable assistants for researchers in the EECP field and likely other medical domains, offering new momentum for scientific progress.

## Supporting information

**S1 File. Raw data and results** - ERNIE Botts 2 and ChatGPT.
(ZIP)

## Author contributions

**Data curation:** Fangwan Huang, Xuanyun Liu.

**Formal analysis:** Fangwan Huang, Xuanyun Liu.

**Methodology:** Shengkun Gai.

**Supervision:** Ryan G. Benton, Glen M. Borchert, Xiuyu Leng.

**Writing – original draft:** Shengkun Gai.

**Writing – review & editing:** Shengkun Gai, Jingshan Huang.

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
