## [Decision Letter · Decision Letter 0]

1 Oct 2024

PONE-D-24-20832Assessing the Feasibility of Large Language Models to Identify TopResearch Priorities in Enhanced External CounterpulsationPLOS ONE

Dear Dr. Leng,

Thank you for submitting your manuscript to PLOS ONE. After careful consideration, we feel that it has merit but does not fully meet PLOS ONE’s publication criteria as it currently stands. Therefore, we invite you to submit a revised version of the manuscript that addresses the points raised during the review process.

We look forward to receiving your revised manuscript.

Kind regards,

Jiwei Tian

Academic Editor

PLOS ONE

Journal Requirements:

1. When submitting your revision, we need you to address these additional requirements. Please ensure that your manuscript meets PLOS ONE's style requirements, including those for file naming. The PLOS ONE style templates can be found at https://journals.plos.org/plosone/s/file?id=wjVg/PLOSOne_formatting_sample_main_body.pdf and https://journals.plos.org/plosone/s/file?id=ba62/PLOSOne_formatting_sample_title_authors_affiliations.pdf 2. We note that your Data Availability Statement is currently as follows: All relevant data are within the manuscript and its Supporting Information files.” Please confirm at this time whether or not your submission contains all raw data required to replicate the results of your study. Authors must share the “minimal data set” for their submission. PLOS defines the minimal data set to consist of the data required to replicate all study findings reported in the article, as well as related metadata and methods (https://journals.plos.org/plosone/s/data-availability#loc-minimal-data-set-definition). For example, authors should submit the following data: - The values behind the means, standard deviations and other measures reported;- The values used to build graphs;- The points extracted from images for analysis. Authors do not need to submit their entire data set if only a portion of the data was used in the reported study. If your submission does not contain these data, please either upload them as Supporting Information files or deposit them to a stable, public repository and provide us with the relevant URLs, DOIs, or accession numbers. For a list of recommended repositories, please see https://journals.plos.org/plosone/s/recommended-repositories. If there are ethical or legal restrictions on sharing a de-identified data set, please explain them in detail (e.g., data contain potentially sensitive information, data are owned by a third-party organization, etc.) and who has imposed them (e.g., an ethics committee). Please also provide contact information for a data access committee, ethics committee, or other institutional body to which data requests may be sent. If data are owned by a third party, please indicate how others may request data access. 3. When completing the data availability statement of the submission form, you indicated that you will make your data available on acceptance. We strongly recommend all authors decide on a data sharing plan before acceptance, as the process can be lengthy and hold up publication timelines. Please note that, though access restrictions are acceptable now, your entire data will need to be made freely accessible if your manuscript is accepted for publication. This policy applies to all data except where public deposition would breach compliance with the protocol approved by your research ethics board. If you are unable to adhere to our open data policy, please kindly revise your statement to explain your reasoning and we will seek the editor's input on an exemption. Please be assured that, once you have provided your new statement, the assessment of your exemption will not hold up the peer review process. 4. PLOS requires an ORCID iD for the corresponding author in Editorial Manager on papers submitted after December 6th, 2016. Please ensure that you have an ORCID iD and that it is validated in Editorial Manager. To do this, go to ‘Update my Information’ (in the upper left-hand corner of the main menu), and click on the Fetch/Validate link next to the ORCID field. This will take you to the ORCID site and allow you to create a new iD or authenticate a pre-existing iD in Editorial Manager.

Additional Editor Comments:

Thank you for submitting your manuscript to PLOS ONE. Enclosed are your reviews. We have completed the review process of the above referenced paper and recommend your paper undergo a Major Revision.

Reviewers' comments:

Reviewer's Responses to Questions

**Comments to the Author**

1. Is the manuscript technically sound, and do the data support the conclusions?

Reviewer #1: Yes

Reviewer #2: Partly

2. Has the statistical analysis been performed appropriately and rigorously? 

Reviewer #1: Yes

Reviewer #2: No

3. Have the authors made all data underlying the findings in their manuscript fully available?

Reviewer #1: Yes

Reviewer #2: No

4. Is the manuscript presented in an intelligible fashion and written in standard English?

Reviewer #1: Yes

Reviewer #2: Yes

5. Review Comments to the Author

Reviewer #1: The paper uses large language models (such as ChatGPT and Ernie Bot) to identify top research priorities in five key topics in the field of EECP: mechanisms, device improvements, cardiovascular applications, neurological

applications, and other applications. It's interesting! To further improve the manuscript, the following suggestions are given:

1、In the paper, some figures in the manuscript are a little blurry, please improve the clarity.

2、Since there are some papers in this topic, the contributions of the manuscript should be better summarized and listed.

3、While the introduction sets the context, a more explicit literature review section could better situate the study within the broader research landscape. I also recommend considering these paper, such as Evaluating the Use of Large Language Model, LESSON Multi-Label Adversarial False Data Injection Attack for Deep Learning Locational Detection, Application Research of Large Language Models in Medicine: Status, Problems, and Future and so on. These references could provide valuable insights into your research.

4、Add a section on the limitations of the work and future work in this paper.

5、The manuscript contains a number of linguistic errors that hinder comprehension. The authors are advised to make careful revisions and improvements.

Reviewer #2: This study tackles an interesting and emerging topic that use powerful LLMs to identify top research priorities in EEC. It is well known that ChatGPT was trained on general text and it is not meant to solve complex deep domain-specific problems. The main conclusion derived in this study is expected. The reviewer is not capable of judging the soundness of the generated research priorities. The ranking from each selected expert is meaningful. However, their aggregation may not as the rankings of different experts may not comparable. Hence, the mean/std in Table 5 may not be very meaningful.

6. PLOS authors have the option to publish the peer review history of their article (what does this mean? ). If published, this will include your full peer review and any attached files.

**Do you want your identity to be public for this peer review?** For information about this choice, including consent withdrawal, please see our Privacy Policy .

Reviewer #1: No

Reviewer #2: No

---

## [Author Response · Author response to Decision Letter 0]

6 Nov 2024

Dear Editor and Reviewer,

On behalf of my co-authors, we thank you very much for giving us an opportunity to revise our manuscript, and we also appreciate reviewers very much for their positive and constructive comments and suggestions on our manuscript entitled “Assessing the Feasibility of Large Language Models to Identify TopResearch Priorities in Enhanced External Counterpulsation” (Manuscript Number: PONE-D-24-20832).

We revised the manuscript according to these comments and suggestions. In general, we have tried our best to revise our manuscript and provide the point-by-point responses. All changes were marked in red using the “Track Changes” function in the revised manuscript. Attached please find our responses to the referees’ comments.

Once again, thank you very much for your comments and suggestions. And we hope that the revised manuscript can be accepted by PLOS ONE. If further revision is necessary, please contact me at: lengxy@mail.sysu.edu.cn

Thank you and best regards.

Sincerely yours,

Xiuyu Leng

---

## [Decision Letter · Decision Letter 1]

4 Feb 2025

PONE-D-24-20832R1Assessing the Feasibility of Large Language Models to Identify TopResearch Priorities in Enhanced External Counterpulsation

PLOS ONE

Dear Dr. Leng,

Thank you for submitting your manuscript to PLOS ONE. After careful consideration, we feel that it has merit but does not fully meet PLOS ONE’s publication criteria as it currently stands. Therefore, we invite you to submit a revised version of the manuscript that addresses the points raised during the review process.

The Reviewer has comment on the availabiliyt of Evidence as mentioned below. 

"It is very important that the authors provide evidence to support their "Expert Evaluation". Please provide the detail as annex to proceed further.

We look forward to receiving your revised manuscript.

Kind regards,

Asim Mehmood

Academic Editor

PLOS ONE

Journal Requirements:

Reviewers' comments:

Reviewer's Responses to Questions

**Comments to the Author**

1. If the authors have adequately addressed your comments raised in a previous round of review and you feel that this manuscript is now acceptable for publication, you may indicate that here to bypass the “Comments to the Author” section, enter your conflict of interest statement in the “Confidential to Editor” section, and submit your "Accept" recommendation.

Reviewer #2: All comments have been addressed

Reviewer #3: All comments have been addressed

2. Is the manuscript technically sound, and do the data support the conclusions?

Reviewer #2: Yes

Reviewer #3: Yes

3. Has the statistical analysis been performed appropriately and rigorously? 

Reviewer #2: N/A

Reviewer #3: N/A

4. Have the authors made all data underlying the findings in their manuscript fully available?

Reviewer #2: No

Reviewer #3: Yes

5. Is the manuscript presented in an intelligible fashion and written in standard English?

Reviewer #2: Yes

Reviewer #3: Yes

6. Review Comments to the Author

Reviewer #2: Please add detailed background information of the panelists in the appendix to support the claim in "Expert Evaluation".

Reviewer #3: The review work carried out in this article was adequate. The work is well structured; the communication with the reader is straightforward and easy to understand; the conclusions are based on the data produced. The references used are adequate. I believe that the paper is in a position to be considered for publication.

7. PLOS authors have the option to publish the peer review history of their article (what does this mean? ). If published, this will include your full peer review and any attached files.

**Do you want your identity to be public for this peer review?** For information about this choice, including consent withdrawal, please see our Privacy Policy .

Reviewer #2: No

Reviewer #3: **Yes: ** Prof. Dr. Marcio Basilio

---

## [Author Response · Author response to Decision Letter 1]

19 Feb 2025

We sincerely appreciate your professional review of our article. Based on your feedback,we have supplemented the expert profiles, which are detailed in the attachment entitled "Curriculum Vitae of Review Experts".Additionally, the raw data and results of the expert evaluation are provided in Appendix "raw data and results "for further reference.

---

## [Editor Report · Decision Letter 2]

25 Feb 2025

Assessing the Feasibility of Large Language Models to Identify TopResearch Priorities in Enhanced External Counterpulsation

PONE-D-24-20832R2

Dear Dr. Xiuyu Leng,

We’re pleased to inform you that your manuscript has been judged scientifically suitable for publication and will be formally accepted for publication once it meets all outstanding technical requirements.

Kind regards,

Asim Mehmood

Academic Editor

PLOS ONE
---

## [Editor Report · Acceptance letter]

PONE-D-24-20832R2

PLOS ONE

Dear Dr. Leng,

I'm pleased to inform you that your manuscript has been deemed suitable for publication in PLOS ONE. Congratulations! Your manuscript is now being handed over to our production team.

Kind regards,

on behalf of

Dr. Asim Mehmood

Academic Editor

PLOS ONE